# Bushmeat Species Identification: Recombinase Polymerase Amplification (RPA) Combined with Lateral Flow (LF) Strip for Identification of Formosan Reeves’ Muntjac (*Muntiacus reevesi micrurus*)

**DOI:** 10.3390/ani11020426

**Published:** 2021-02-07

**Authors:** Yun-Hsiu Hsu, Wei-Cheng Yang, Kun-Wei Chan

**Affiliations:** 1Department of Veterinary Medicine, College of Veterinary Medicine, National Chiayi University, Chiayi 60054, Taiwan; asho9937@gmail.com; 2Department of Veterinary Medicine, School of Veterinary Medicine, National Taiwan University, Taipei 10617, Taiwan

**Keywords:** bushmeat, Formosan Reeves’ muntjac, recombinase polymerase amplification, lateral flow strip

## Abstract

**Simple Summary:**

Illegal hunting of wild animals and the consumption of bushmeat are recognized not only as a threat to biodiversity, but also as a risk for transmitting zoonotic diseases. Illegal sales of meat products from Formosan Reeves’ muntjac (*Muntiacus reevesi micrurus*) is a growing issue in Taiwan, bringing forth the demand for a fast and cost-effective technique for meat species identification. In this study, a new recombinase polymerase amplification combined with a lateral flow strip to identify Formosan Reeves’ muntjac in meat products was described. This method only requires minimal sample preparation and an isothermal heating process. The result can be interpreted by the naked eye within 30 min. The system we designed efficiently detected a variety of meat products, and no cross-reactions were observed with other animal species. This simple assay provides a sensitive and specific method to identify bushmeat sources in various meat products, which holds the potential for on-field application in the future.

**Abstract:**

The identification of animal species of meat in meat products is of great concern for various reasons, such as public health, religious beliefs, food allergies, legal perspectives, and bushmeat control. In this study, we developed a new technique to identify Formosan Reeves’ muntjac in meat using recombinase polymerase amplification (RPA) in combination with a lateral flow (LF) strip. The DNA extracted from a piece of Formosan Reeves’ muntjac meat was amplified by a pair of specific primers based on its mitochondrial cytochrome b gene for 10 min at a constant temperature ranging from 30 to 45 °C using RPA. Using the specific probe added to the RPA reaction system, the amplified products were visualized on the LF strip within 5 min. The total operating time from quick DNA extraction to visualizing the result was approximately 30 min. The RPA-LF system we designed was efficient when using boiled, pan-fried, roasted, stir-fried, or stewed samples. The advantages of simple operation, speediness, and cost-effectiveness make our RPA-LF method a promising molecular detection tool for meat species identification of either raw or variously cooked Formosan Reeves’ muntjac meat. It is also possible to apply this method to identify the meat of other wildlife sources.

## 1. Introduction

Formosan Reeves’ muntjac (*Muntiacus reevesi micrurus*), a subspecies of *Muntiacus reevesi*, classified and protected as a rare and valuable species in Taiwan before 2019, is now widely distributed on the island. Due to its increasing population, the Taiwan government relaxed the restriction of the Wildlife Conservation Act, allowing Taiwanese Indigenous people to hunt Formosan Reeves’ muntjacs for cultural or ritual purposes only. However, Formosan Reeves’ muntjacs are sometimes sold to restaurants for business. The suspected meats sold in restaurants were delivered to a professional laboratory for source identification, but this time-consuming analysis usually causes an economic loss to the restaurant if the distrained meat products were eventually defined as legal meats. This issue demands a fast, simple, and cost-effective identification technique for wildlife conservation officers to identify meat sources on-site.

A variety of analytic methods have been applied to differentiate animal species of meat, including ELISA [1,2,3,4], immunochromatography [5], and Raman spectroscopy [6,7]. Currently, PCR-based techniques, such as species-specific PCR [8,9,10,11,12], restriction fragment length polymorphism (RFLP) [13,14], and real-time PCR [15,16,17], are widely used. However, these assays are time-consuming, laborious, and require professional equipment. Recently, recombinase polymerase amplification (RPA), a novel nucleic acid amplification method, has been established [18]. RPA can be performed at temperature settings ranging from 25 to 42 °C and requires only 1–10 copies of target DNA. The reaction can be completed in 5 to 20 min and visualized by either gel electrophoresis, nucleic acid dye, or a lateral flow (LF) strip [19,20]. RPA is also resistant to common PCR inhibitors, and the reaction can be performed with heated, lysed, or crude DNA extract [21,22,23,24,25]. Recently, diagnostic techniques using RPA have successfully been developed for various pathogens, including viruses [23,25], bacteria [26,27], parasites [21,24], and fungi [28].

To our knowledge, RPA has not been applied to the identification of wild animal meat. In this study, we describe the first development of the RPA-LF assay to detect Formosan Reeves’ muntjac DNA (*frm*RPA-LF). The cytochrome b gene was chosen as the target for the *frm*RPA-LF. We also evaluated the difference in the performance of the *frm*RPA-LF between crude DNA extract and purified DNA extract, as well as among DNA samples extracted from meat processed by different cooking methods.

## 2. Materials and Methods

### 2.1. Sample Collection

All wild animal meat samples, including Formosan Reeves’ muntjac, Formosan serow, masked palm civet, Formosan macaque, and Formosan pangolin were provided by the Forestry Bureau of the Council of Agriculture, Executive Yuan, and the Institute of Wildlife Conservation at the National Pingtung University of Science and Technology. Other meat samples, such as pork, beef, chicken, salmon, ostrich, lamb, tilapia, and rabbit, were obtained from local markets. All of the meat samples were frozen and stored at −20 °C until used.

### 2.2. Preparation of Cooked Meat

The Formosan Reeves’ muntjac meat was diced into 7 pieces of 3 cm × 3 cm × 1 cm, and then cooked until ready to consume. Each piece of meat was cooked in one of the following 7 different cooking methods: boiling, roasting, pan-frying, stewing, stir-frying with soy sauce, stir-frying with barbecue sauce, and stir-frying with sesame oil. For boiling, a diced sample was boiled for 5 min in 1000 mL of water with half a teaspoon of salt. A convection oven was used to roast the Formosan Reeves’ muntjac meat, mixed with garlic, pepper, and salt at 180 °C for 10 min. For pan-frying, the diced sample was fried in a stainless-steel pan at 170 °C for 5 min. For stewing, the sample was braised in a stainless-steel cooking pot with 500 mL of water, 100 mL of red wine and half a teaspoon of salt for 20 min. Three other pieces of meat samples were stir-fried separately for 10 min in soy sauce, barbecue sauce, or sesame oil. All the cooked meat samples were rinsed with water before DNA extraction procedures.

### 2.3. Preparation of Purified Genomic DNA

Purified genomic DNA was extracted from the meat samples using the DNeasy^®^ Blood & Tissue Kit (QIAGEN, Valencia, CA, USA). Animal muscle tissue, weighing approximately 10–15 mg, was first ground in a 180 μL Buffer ATL. Then, Proteinase K was added into the mixture, preceding incubation at 56 °C for approximately one hour until complete digestion of the tissue. The following extraction procedures were performed according to the manufacturer’s protocols: a total volume of 200 μL of DNA mixture was eluted and 5 μL of the product was then used in the Qubit 2.0 Fluorometer (Invitrogen, Carlsbad, CA, USA) to estimate the quantity of genomic DNA.

### 2.4. Preparation of Crude Genomic DNA

To shorten the diagnostic time, a single-step extraction technique of UniversAll Tissue Extraction Buffer (Yeastern Biotec, Taipei, Taiwan) was tested for its applicability. For each of the 8 cooked and uncooked Formosan Reeves’ muntjac samples, a 2 mm cube was placed into a 1.5 mL Eppendorf tube and mixed with 100 μL of the extraction buffer. The cube of muscle tissue was then ground with a disposable polypropylene pellet pestle. The pestle was gently rotated back and forth manually for 30 s until the meat sample was well-crushed. The genomic DNA was obtained and ready to use after incubating at room temperature for 10 min. Although the required temperature for UniversAll Tissue Extraction Buffer, according to the instructions, was 37 °C, the samples we extracted under room temperature in this study yielded a sufficient amount of DNA for the following tests (see Table 1). The grinding procedure may facilitate the DNA extraction process and make the incubation at room temperature work as efficiently as incubating at 95 °C.

### 2.5. Primer Design

Full cytochrome b sequences of Formosan Reeves’ muntjac (accession number: EF035447) and common food animals were downloaded from GenBank and aligned on the software DNAStar (DNASTAR, Madison, WI, USA). Primers were designed using the software Primer3 Input (http://bioinfo.ut.ee/primer3/) and NetPrimer (http://www.premierbiosoft.com/netprimer/) for initial primer screening. According to the TwistAmp DNA Amplification Kits Assay Design Manual, the primer size is best between 30 and 36 nucleotides and its GC content between 20% and 70%. The range of the primer Tm value is set between 50 to 100. Primer dimers, hairpins, and self-annealing are prone to cause artifacts and thus should be avoided. Reverse primers were labeled with biotin for the lateral flow strip analysis. The expected size of the amplification product is best between 100 and 200 bp, in order to shorten the amplification time.

### 2.6. Probe Design

For the utilization of the lateral flow detection system, we designed a labeling probe. The 5′ end of the probe was labeled with carboxyfluorescein (FAM), while the 3′ end was blocked by a three-carbon (C3) spacer. Then, the THF (tetrahydrofuran) site was placed 30 bp from the 5′ end of the probe, replacing one nucleotide. At least 15 bp should be added to the 3′ end after the THF residue. There is no fixed instruction describing the best location for a probe, but the possibility of dimer artifacts should still be avoided.

### 2.7. Optimization of frmRPA-LF

The *frm*RPA-LF was performed with the TwistAmp nfo Kit (TwistDx, Cambridge, UK). The reaction mixture consisted of 2.1 μL of both *frm*-forward and *frm*-reverse primers (final concentration: 420 nM), 0.6 μL of the *frm*-probe (final concentration: 120 nM), 2.5 μL of magnesium acetate (MgAc; final concentration: 14 nM), 29.5 μL of the rehydration buffer, 10 μL of the purified Formosan Reeves’ muntjac genomic DNA (concentration of 27.7 ng/μL), and 3.2 μL of nuclease-free water. Finally, the nfo RPA freeze-dried pellet was added to the reaction, followed by a brief vortex and spin. To optimize the reaction temperature and time, the amplification reaction mixture was incubated in a Lifepro Gradient Thermal Cycler (Bioer Technology, Hang, China) at different reaction temperatures (25, 30, 35, 40, 45, and 50 °C) for 20 min. Different amplification durations (5, 10, 15, 20, 25, and 30 min) were tested at 37 °C, the temperature instructed by the instruction manual of the TwistAmp nfo Kit (TwistDx, Cambridge, UK).

### 2.8. Lateral Flow Analysis

Lateral flow assays were performed following the instructions provided by the TwistAmp nfo Kit (TwistDx, Cambridge, UK). Briefly, 2 μL of the amplified products were diluted in 98 μL of the HybriDetect Assay Buffer (Milenia Biotec, GieBen, Germany). After properly mixing by pipetting, 10 μL of the diluted products were loaded on the sample pads of the Milenia HybriDetect 1 strips (Milenia Biotec, Giessen, Germany). The strips were directly placed into the HybriDetect Assay Buffer until the whole strip fully absorbed the buffer. Finally, the results were visualized by the naked eye. The reaction was considered positive when both the sample and control lines were visible. If only the control line was visible, the test was interpreted as negative.

### 2.9. Cross-Reactivity Analysis

To evaluate the specificity of the *frm*RPA-LF, a cross-reaction test was performed using 10 μL of the genomic DNA purified from various wild or food animals, including Formosan Reeves’ muntjac (concentration of 27.7 ng/μL), Formosan serow (concentration of 62.0 ng/μL), masked palm civet (concentration of 5.59 ng/μL), Formosan macaque (concentration of 24.63 ng/μL), Formosan pangolin (concentration of 9.11 ng/μL), pig (concentration of 2.24 ng/μL), cow (concentration of 4.52 ng/μL), chicken (concentration of 3.22 ng/μL), salmon (concentration of 8.78 ng/μL), ostrich (concentration of 14.7 ng/μL), goat (concentration of 2.46 ng/μL), rabbit (concentration of 33.2 ng/μL), and tilapia (concentration of 9.16 ng/μL). For negative controls, nuclease-free water was used instead of genomic DNA.

### 2.10. Detection Limit

The detection limit for the meat cooked in different ways was further tested and assessed. In addition to frozen raw meat, 7 common Taiwanese cooking methods including boiling, pan-frying, roasting, stewing, and stir-frying in soy sauce/sesame oil/barbecue sauce were chosen for sensitivity estimation. Besides cooking methods, the difference between purified and crude DNA extracted from cooked and raw meats was also examined to estimate the detection limit of the *frm*RPA-LF. The concentration of each extracted genomic DNA used for the detection limit estimation are listed in Table 1.

## 3. Results

### 3.1. Primer and Probe Selection

As a preliminary experiment, three sets of primer pairs targeting a unique region on mitochondrial cytochrome b gene were initially designed and tested without probes (data not shown). The primer pair we selected to use for this study was designed to yield an amplified product of 193 bp in length. A hybridization probe designed for the LF strip gave a final primer–probe combination product of 128 bp. The primers and probe used in our study are shown in Table 2 and Figure 1.

### 3.2. Optimization of frmRPA-LF Reaction

The assay conditions were optimized by performing the *frm*RPA-LF reaction at various incubation temperatures and amplification times. As shown in Figure 2A, after 20 min of incubation, the temperature settings ranging from 30 to 45 °C gave visually positive bands on the LF strip, although a slightly fainter band was observed at 30 °C. The results suggest that the *frm*RPA-LF reaction would not require strict temperature control, although the temperature settings at 30 °C and lower might not lead to optimal reaction, compared to the settings at 35 to 45 °C. Furthermore, optimization of the reaction time of the *frm*RPA-LF was carried out at a constant temperature of 37 °C by varying the incubation time from 5 to 30 min. As shown in Figure 2B, although the positive reaction was visible after 5 min of incubation, amplification for 10 min and longer may ensure a better quality of the LF strip signal.

### 3.3. Cross-Reactivity Analysis

To evaluate the specificity of the *frm*RPA-LF assay, cross-reactions were tested using the purified genomic DNAs of both wild and food animals. Only the Formosan Reeves’ muntjac strip displayed a positive reaction, and no such signals were observed on the other species (Figure 3).

### 3.4. Detection Limit

The influences of different DNA extraction methods and meat preparations on our *frm*RPA-LF were evaluated using purified and crude DNA extracts from variously cooked and raw Formosan Reeves’ muntjac meat. The DNA concentrations of both purified (Figure 4A) and crude extracts (Figure 4B) are shown in Figure 4. DNA extracted from Formosan Reeves’ muntjac meat gave positive results regardless of the type of meat preparations, from raw to all different cooking methods. The results indicate that our *frm*RPA-LF may be able to detect DNA extracted from both raw and cooked meat and would not be affected by spices, sauce, oil, or different cooking methods.

## 4. Discussion

Illegal bushmeat consumption may lead to the transmission of zoonotic diseases to humans and threaten wildlife biodiversity [29]. The Taiwan Wildlife Conservation Act prohibits hunting wild animals without authorized permission. Still, wildlife is often hunted and sold to restaurants illegally, giving rise to the demand of an available on-site diagnostic tool to identify meat species in food, quickly and accurately. In this study, we demonstrated a new technique of the isothermal RPA in combination with a lateral flow detection to identify Formosan Reeves’ muntjac meat in raw meat and various meat preparations.

A variety of techniques analyzing protein composition have been utilized to identify meat origin [1,5,6,30]. However, heat and pressure processing may cause protein denaturation, posing problems of possible misdiagnosis [31]. Antibodies against heat-stable biomarkers have also been developed for meat species detection and are available as an option; however, these antisera are known to show cross-reaction among different mammalian species [32]. DNA-based detection methods provide a reliable choice for meat species identification, due to DNA’s heat stability and abundance in tissue [33]. We chose the mitochondrial cytochrome b DNA sequence as the target gene because it has a couple of advantages over genomic DNA sequences. First, mitochondrial DNA is present in as many as 2500 copies in each cell, making it a rich amplifiable source of gene detection. Second, the variation of cytochrome b sequence also facilitates species discrimination [34].

When developing a new *frm*RPA-LF technique, the top priority was to prevent cross-reaction with other meat species from happening. Therefore, we aligned the cytochrome b sequences of food animals and wild animals, often sold illegally in restaurants, in order to determine a unique target region for the *frm*RPA-LF assay. The second priority was to optimize the reaction, so we analyzed the optimal temperature and time. The optimal amplification temperature was evaluated using six constant temperature settings. Positive LF signals were observed when performing the *frm*RPA-LF at temperatures ranging from 30 to 45 °C (Figure 2A). However, the positive reaction when amplified at 30 °C was weaker in comparison to higher temperature settings. This may be due to the decreased RPA enzyme kinetics caused by the low temperature, which can be compensated with a longer reaction time [35]. The broad range of incubation temperatures of the *frm*RPA-LF makes it possible to be applied on-site at room temperature, in a water bath, or with body heat. This advantage also makes RPA superior to traditional polymerase chain reaction (PCR) methods because it eliminates the need for a time-consuming thermocycling process. Recent publications have reported that isothermal nucleic acid amplification methods such as helicase-dependent amplification and loop-mediated isothermal amplification may work at a constant temperature ranging from 60 to 65 °C [36,37]. Compared to the high temperature in such isothermal amplification methods, RPA can be operated at a lower constant temperature and does not require strict temperature control. Both the advantages of less energy requirement and stable amplification at different temperature settings facilitate the field applicability of *frm*RPA-LF. In the present study, the time required for amplification to yield a detectable level of positive reaction on the LF strip was 5 min (Figure 2B), which is significantly faster than the recommended reaction time (20 min) by the TwistAmp^®^ nfo Kit Quick Guide. RPA incubation times for 5 min or even less have also been reported in previous studies [38,39,40]. The length of incubation time might be associated with the initial number of DNA copies [20]. Considering that there are thousands of copies of mitochondrial DNA per cell [41], the number of DNA copies might also contribute to the highly efficient amplification of *frm*RPA-LF. Ten minutes of amplification time was decided in order for the *frm*RPA-LF assay to compensate for the low reaction rate caused by the low temperature. After optimizing the reaction conditions, the specificity of our *frm*RPA-LF was evaluated using meat samples from five wild animals and eight food animals. The results from the cross-reaction assay showed that our *frm*RPA-LF method detected specifically the meat samples from Formosan Reeves’ muntjac, and no cross-reaction with other species was observed (Figure 3).

To examine the detection limit, the *frm*RPA-LF assay was performed using raw meat and variously cooked meat samples. As Figure 4 illustrates, cooking methods including boiling, stir-frying with sauces and sesame oil, pan-frying, roasting, and stewing did not affect the detection of DNAs extracted from Formosan Reeves’ muntjac meat. This indicates that both seasoning and heating before DNA extraction from the meat sample would not affect the amplification of the *frm*RPA-LF reaction. Some studies also reported that pork, chicken, lamb, and goat cooked at 120 °C can be identified by PCR, with the exception of horse meat [42,43,44]. In 2006, Arasan and his colleagues tested the influences of different cooking methods including boiling, roasting, pressure cooking, and pan-frying on the accuracy of PCR. The animal species of origin of all the meat samples were successfully determined, with the exception of charred meat samples. The charred meat failed the PCR reaction presumably due to DNA degeneration by excessive heating; thus, it was suggested to avoid sampling the charred portions. The authors also compared the influence of different sizes of PCR products on the results and concluded that smaller-sized products are superior to the larger ones, since DNA can be broken down easily during the cooking process at a high temperature [45]. In our *frm*RPA-LF design, the sizes of the first amplification products (193 bp) and primer–probe hybridization products (128 bp) were both below 200 bp. This not only shortened the amplification time, but also avoided the misdiagnosis due to DNA destruction. Moreover, the results from the evaluation of efficiency of crude DNA extracts showed that crude DNA extracts worked as well as purified DNA (Figure 4b). The use of crude DNA extracts can simplify the *frm*RPA-LF assay and reduce the total cost.

## 5. Conclusions

We successfully developed a novel *frm*RPA-LF assay to identify Formosan Reeves’ muntjac meat in raw and cooked meat samples. It is a fast, simple, sensitive, cost-effective, precise, and potentially applicable technique. The *frm*RPA-LF only requires minimal sample preparation and an isothermal heating process. Within 30 min, the results can be interpreted by the naked eye. Several cooking methods commonly used in Taiwan were tested and none of them affected the efficiency of the *frm*RPA-LF. The isothermal amplification temperature and short reaction time also make the *frm*RPA-LF a good candidate for a point-of-care diagnostic method. In a future study, portable heating platforms will be designed for *frm*RPA-LF to make it more applicable for field use.

## Figures and Tables

**Figure 1 animals-11-00426-f001:**
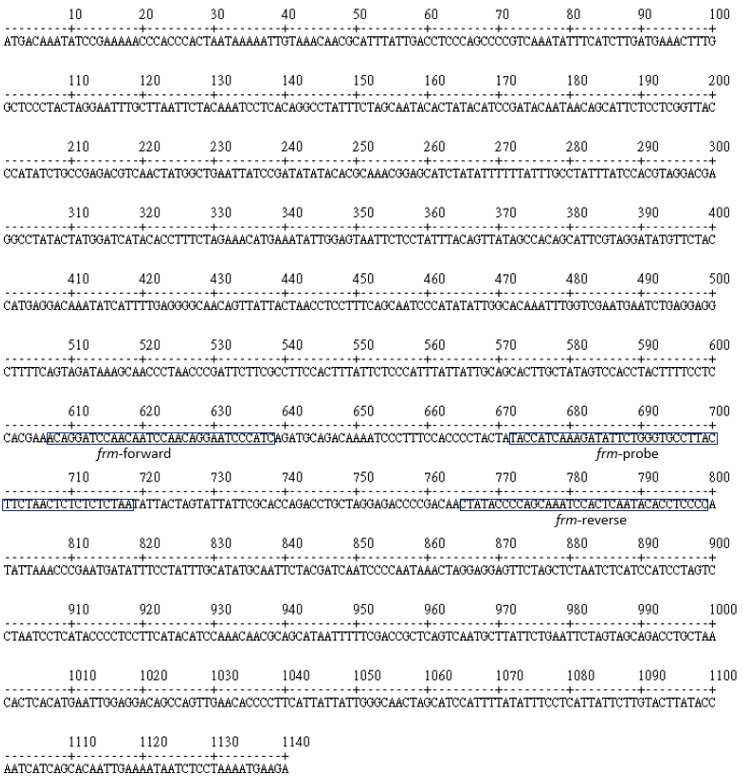
The locations of the primers and probe on the mitochondrial cytochrome b gene sequence of Formosan Reeves’ muntjac.

**Figure 2 animals-11-00426-f002:**
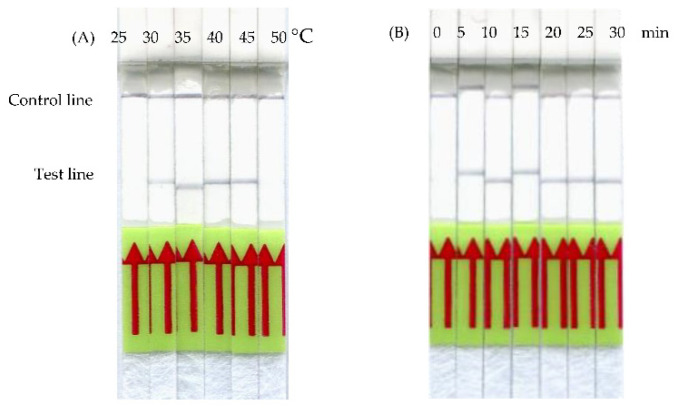
Optimization of the *frm*RPA-LF incubation time and temperature. (**A**) Amplification for 20 min at a constant temperature ranging from 35 to 45 °C yielded distinct positive reactions. However, the positive reaction at 30 °C was slightly weaker. (**B**) After 5 min of incubation at 37 °C, clear positive signals started to show up on lateral flow (LF) strips.

**Figure 3 animals-11-00426-f003:**
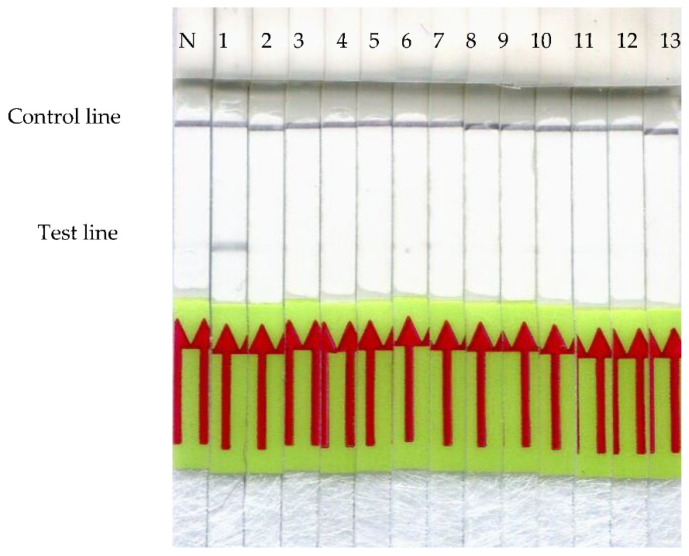
Cross-reaction of the *frm*RPA-LF. Strips 1–13 indicate Formosan Reeves’ muntjac, Formosan serow, masked palm civet, Formosan macaque, Formosan pangolin, pig, cow, chicken, salmon, ostrich, goat, rabbit, and tilapia. N is the result of nuclease-free water instead of DNA. No positive reaction was displayed on the amplification results of other animal species.

**Figure 4 animals-11-00426-f004:**
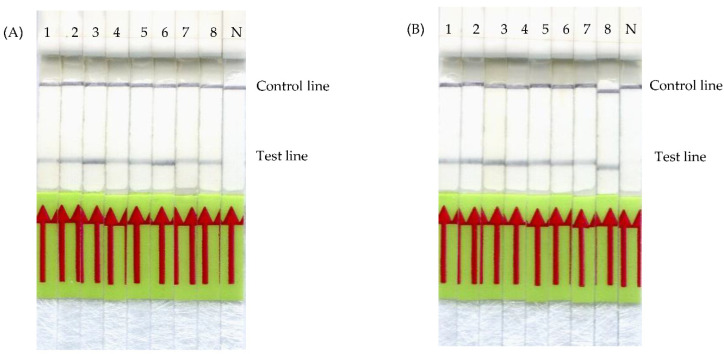
Detection limit of the *frm*RPA-LF. (**A**) *frm*RPA-LF using purified DNA and (**B**) crude DNA extracted from variously-cooked Formosan Reeves’ muntjac meat. Strips 1–8 indicate the results from raw meat, meat stir-fried in soy sauce, boiled meat, pan-fried meat, roasted meat, meat stir-fried in sesame oil, meat stir-fried in barbecue sauce, and stewed meat. N is the result of nuclease-free water instead of DNA. Successful amplification and clear positive results were obtained from all the purified and crude DNA extracts regardless of the type of meat samples, whether the meat was raw or cooked in various ways.

**Table 1 animals-11-00426-t001:** DNA concentration (ng/μL) of different meat samples from Formosan Reeves’ muntjac, extracted using DNeasy^®^ Blood & Tissue Kit and UniversAll Tissue Extraction Buffer.

Meat Type	DNeasy^®^ Blood & Tissue Kit	UniversAll Tissue Extraction Buffer
Raw	27.7	4.84
Stir-fried in soy sauce	46.4	3.02
Boiled	80.2	5.66
Pan-fried	40.4	4.06
Roasted	44.4	4.64
Stir-fried in sesame oil	46.4	5.84
Stir-fried in barbecue sauce	58.6	2.42
Stewed	45.2	1.56

**Table 2 animals-11-00426-t002:** Primers and probe used in this study.

Primer	Sequence (5′-3′)	Gene Location (EF035447)
*frm*-forward	5′-ACAGGATCCAACAATCCAACAGGAATCCCATC-3′	607–638
*frm*-reverse	5′-Biotin-GGGGAGGTGTATTGAGTGGATTTGCTGGGGTATAG-3′	765–799
*frm*-probe	5′-FAM-TACCATCAAAGATATTCTGGGTGCCTTACTTCT (THF) AACTCTCTCTCTAA C3 spacer-3′	672–718

THF: tetrahydrofuran.

## Data Availability

Data is contained within the article.

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
