# Peer review of "Bushmeat Species Identification: Recombinase Polymerase Amplification (RPA) Combined with Lateral Flow (LF) Strip for Identification of Formosan Reeves’ Muntjac (*Muntiacus reevesi micrurus*)"

_animals, 2021, doi:10.3390/ani11020426_

Round 1

Reviewer 1 Report

The reviewer finds the author responses and manuscript changes sound. The revised manuscript is indeed improved. 

Reviewer 2 Report

Dear authors,

Thanks for considering my comments and editing.

Wishes

This manuscript is a resubmission of an earlier submission. The following is a list of the peer review reports and author responses from that submission.

Round 1

Reviewer 1 Report

In this work, the authors report a new assay based on recombinase polymerase amplification (RPA) and lateral flow (LF) detection to identify Formosan Reeves’ Muntjac meat in raw meat and cooked meat samples. The authors developed their “frmRPA-LF” assay by designing and screening 3 pairs of primers that specifically target the mitochondrial cytochrome b gene of Formosan Reeves’ Muntjac, and then optimized the reaction temperature of RPA and the detection time of lateral flow strip. The authors checked the assay specificity using purified DNA from 12 other animals. Finally, the authors detected both purified DNA and crude DNA of Formosan Reeves’ Muntjac from both raw meat and cooked meat samples. The authors believe that the isothermal amplification temperature and short reaction time make their frmRPA-LF a good candidate for identifying Formosan Reeves’ Muntjac meat in the field.

The strength of this work, as the authors point out, centers on the first demonstration of an RPA-LF assay for the identification of Formosan Reeves’ Muntjac meat in raw meat and cooked meat samples. On the other hand, the manuscript in its current form still must be improved. The reviewer believes that the authors still must determine the analytical sensitivity of their assay and specify missing experimental details. The authors should also provide stronger rationales for the application of their assay and some of the experiments. The major and minor comments are listed below.   

Major Comments:

  1. For assay development, it is convention to determine the analytical sensitivity of the assay. The authors are asked to test their assay against a titration series of Formosan Reeves’ Muntjac DNA. This can be done with either purified DNA or synthesized DNA. Based on the results, the authors are then asked to determine the limit of detection based on the amount of input DNA.
  2. The authors must specify the amount of DNA (weight or copy number) that was used for the results shown in Figures 1, 2, and 3.
  3. The reviewer would like the authors to show the primers and the probes alignment to the mitochondrial cytochrome b gene sequence of Formosan Reeves’ Muntjac as an additional figure. The authors are also asked to include the GenBank accession number for the sequence.
  4. On lines 44 – 46, the authors state that “However, these wild animals are sometimes sold to restaurants for business. This issue demands a fast, simple, and cost-effective identification technique for meat species identification.” There is quite a bit of leap in reasoning between the two sentences, and the reviewer would like the authors to close the gap between the two sentences and elaborate more on the rationale of their work. Likewise, on line 271, the authors opined that their assay can be a candidate for a point-of-care diagnostic method, but the reviewer is unsure why meat species identification should be done at the point of care. The authors are asked to improve the rationale here as well.
  5. The reviewer was intrigued by the tests using various meat samples cooked with different methods. But the reviewer was unsure why that was necessary. The reviewer can understand that cooking can denature proteins and therefore cause immunoassays to fail, which provides an example for the superiority of DNA-based detection. So the reviewer could see the rationale of testing both raw meat and cooked meat. The reviewer can also understand that charred meats may not yield quality DNA for analysis. So, if the authors’ rationale is to evaluate how much charred meat samples would affect their assay, that is understandable. But if the focus was on either seasoning or cooking methods, then the authors need to provide more explanation.
  6. The reviewer asks the authors to address two questions related to methods, both in the preparation of crude DNA section (Section 2.4). First, the authors stated that “the cube of muscle tissue was then ground with disposable polypropylene pellet.” The reviewer asks the authors to provide thorough details about this process. Second, the authors stated that “The genomic DNA was obtained and ready to use after incubation at room temperature for 10 min.” This step seems to deviate from the product information of UniversAll Tissue Extraction Buffer, which requires 10 min incubation at 95 degrees C. If the authors were able to improve upon the vendor recommended procedure, the reviewer asks the authors to briefly note the difference and explain what facilitated this change.

Minor Comments:

  1. The manuscript in its current form still requires additional editorial feedback. Several little errors could be seen by the reviewer just in the Simple Summary.
  2. On line 42, the reviewer would like the authors to clarify if they actually mean “decreasing” population of Formosan Reeves’ Muntjac instead of “increasing”.
  3. On lines 212 – 214, the reviewer asks the authors to provide references for the statements: “First, cytochrome b sequence is present within every mitochondrion, making it a rich amplifiable source of gene detection. Second, the variation of cytochrome b sequence also facilitates the species discrimination.”

Reviewer 2 Report

The authors develop recombinase polymerase amplification (RPA) combined with lateral flow (LF) strip for identification of Formosan Reeves’ Muntjac (Muntiacus reevesi micrurus). Introduction is well written while there are major concerns within the methodology and results sections:

  1. Figure 1. a) The authors use different temperature for 20 minutes while b) they apply the test at 37c, why? Why the authors did not use on the pre-optimized temperatures?
  2. Why the authors apply only RPA followed by LF without any comparison for real time PCR regarding the sensitivity? It is very important to compare the RPA data to real time PCR results. Also, the authors should run the RPA amplified products on gel to see the primers dimers effect.
  3. Why the authors did not show the RPA results?
  4. The control line within the LF, which host genes it detect?
  5. Again in Figure 3, the authors should present the RPA data either amplification graphs and or gel electrophoresis.
  6. The authors use different DNA extraction methods while they did not present any relevant data regarding them within the results sections.

Overall, the manuscript need grammar fine tuning.
